# A Limited-View CT Reconstruction Framework Based on Hybrid Domains and Spatial Correlation

**DOI:** 10.3390/s22041446

**Published:** 2022-02-13

**Authors:** Ken Deng, Chang Sun, Wuxuan Gong, Yitong Liu, Hongwen Yang

**Affiliations:** Institute of Wireless Theories and Technologies Laboratory, Beijing University of Posts and Telecommunications, Haidian, Beijing 100876, China; arieldeng@bupt.edu.cn (K.D.); sc1998@bupt.edu.cn (C.S.); gongwuxuan@bupt.edu.cn (W.G.); yanghong@bupt.edu.cn (H.Y.)

**Keywords:** CT image reconstruction, low dose protocol, adversarial autoencoder, deep learning, hybrid domain, spatial correlation, inverse problems

## Abstract

Limited-view Computed Tomography (CT) can be used to efficaciously reduce radiation dose in clinical diagnosis, it is also adopted when encountering inevitable mechanical and physical limitation in industrial inspection. Nevertheless, limited-view CT leads to severe artifacts in its imaging, which turns out to be a major issue in the low dose protocol. Thus, how to exploit the limited prior information to obtain high-quality CT images becomes a crucial issue. We notice that almost all existing methods solely focus on a single CT image while neglecting the solid fact that, the scanned objects are always highly spatially correlated. Consequently, there lies bountiful spatial information between these acquired consecutive CT images, which is still largely left to be exploited. In this paper, we propose a novel hybrid-domain structure composed of fully convolutional networks that groundbreakingly explores the three-dimensional neighborhood and works in a “coarse-to-fine” manner. We first conduct data completion in the Radon domain, and transform the obtained full-view Radon data into images through FBP. Subsequently, we employ the spatial correlation between continuous CT images to productively restore them and then refine the image texture to finally receive the ideal high-quality CT images, achieving PSNR of 40.209 and SSIM of 0.943. Besides, unlike other current limited-view CT reconstruction methods, we adopt FBP (and implement it on GPUs) instead of SART-TV to significantly accelerate the overall procedure and realize it in an end-to-end manner.

## 1. Introduction

Computed Tomography (CT) [1] is diffusely known as an approach to exhibit precise details inside the scanned object [2], thus is applied to a wide range of applications including clinical diagnosis, industrial inspection, material science and biomedicine [3,4]. In addition, the raging epidemic caused by the Corona Virus Disease 2019 (COVID-19) has made CT known to the public as an efficacious auxiliary technology. Nevertheless, the associated x-ray radiation dose brings potential risk of cancers [5], which has drawn wide attention. Consequently, the demand of radiation dose reduction is becoming more and more acute under the principle of ALARA (as low as reasonably achievable) [6,7,8,9,10].

Generally, Low-dose Computed Tomography (LDCT) can be realized through two strategies including current (or voltage) reduction [11,12] and projection reduction [13,14,15]. The first strategy aims to lower the x-ray exposure in each view, while it greatly suffers from the increased noise in projections. Although the second strategy can avoid the above problem and realize the additional benefit of accelerated scanning and calculation, it gives rise to severe image quality deterioration of increased artifacts due to its lack of projections. In this paper, we will focus on obtaining high-quality CT images from limited-view CT with inadequate scanning angle.

Researchers have proposed various CT image reconstruction algorithms in the past few decades, but when it comes to LDCT reconstruction, the problem becomes challenging. Traditional analytical reconstruction algorithms, such as FBP [16], have high requirements for data integrity. When the radiation dose is reduced, artifacts in reconstructed images will increase rapidly [17]. Compared with analytical reconstruction algorithms, iterative reconstruction algorithms can obtain better performance, while suffering from higher complexity. Model-based iterative reconstruction (MBIR) algorithm [18], combines the modeling of some key parameters to perform high-quality reconstruction of LDCT. Using image priors in MBIR can effectively improve the image reconstruction quality of LDCT scans [14,19], while still have the high computational complexity.

In addition, diverse regularization methods have played a crucial role in CT reconstruction, which is a typical inverse problem. The most prevailing regularization method is the total variation (TV) method [20]. In the light of TV, researchers came up with more reconstruction methods, such as TV-POCS [21], TGV [22] and SART-TV [13] which was proposed on the basis of SART [23]. Those algorithms can suppress image artifacts to a certain extent so as to improve imaging quality. In addition, dictionary learning is often used as a regularizer in MBIR algorithms [24,25,26,27], and multiple dictionaries are beneficial to reducing artifacts caused by limited-view CT reconstruction.

With the development of computing power, deep learning-based methods [28,29,30,31,32,33,34] have been applied to the restoration of LDCT reconstructed images in recent years. The methods can be roughly divided into the below three categories.

Image inpainting algorithms usually reconstruct the damaged Radon data into the damaged image with artifacts through regular methods, such as FBP, then reduce the artifacts and noises in the image domain. Lots of researchers are currently using convolutional neural network (CNN) and deep learning architecture to perform this procedure [4,35,36,37,38,39,40,41,42,43,44]. Zhang et al. [35] proposed a data-driven learning method based on deep CNN. RED-CNN [4] combines the autoencoder, deconvolutional network and shortcut connections into the residual encoder-decoder CNN for LDCT imaging. Kang et al. [36] applied deep CNN to the wavelet transform coefficients of LDCT images, used directional wavelet transform to extract the directional component of artifacts. Wang et al. [39] developed a limited-angle translational CT (TCT) image reconstruction algorithm based on U-Net [40]. Since Goodfellow et al. proposed Generative Adversarial Nets (GAN) [42] in 2014, GAN has been widely used in various image processing tasks, including the post-processing of CT images. Xie et al. [43] proposed an end-to-end conditional GAN with joint loss function, which can effectively remove artifacts.

Sinogram inpainting algorithms firstly restore the missing part in the Radon domain, then reconstruct it into the image domain to get the final result [45,46,47,48,49]. Li et al. [45] proposed an effective GAN-based repairing method named patch-GAN, which trains the network to learn the data distribution of the sinogram to restore the missing sinogram data. In another paper [46], Li et al. proposed SI-GAN on the basis of [37], using a joint loss function combining the Radon domain and the image domain to repair “ultra-limited-angle” sinogram. In 2019, Dai et al. [47] proposed a limited-view cone-beam CT reconstruction algorithm. It slices the cone-beam projection data into the sequence of two-dimensional images, uses an autoencoder network to estimate the missing part, then stack them in order and finally use FDK [50] for three-dimensional reconstruction. Anirudh et al. [48] transformed the missing sinogram into a latent space through a fully convolutional one-dimensional CNN, then used GAN to complement the missing part. Dai et al. [49] calculated the geometric image moment based on the projection-geometric moment transformation of the known Radon data, then estimated the projection-geometric moment transformation of the unknown Radon data based on the geometric image moment.

Sinogram inpainting and image refining algorithms firstly restore the missing part in the Radon domain, then reconstruct the full-view Radon data into the image domain so as to finely repair the image to obtain higher quality [51,52,53,54,55]. In 2017, Hammernik et al. [51] proposed a two-stage deep learning architecture, they first learn the compensation weights that account for the missing data in the projection domain, then they formulate the image restoration problem as a variational network to eliminate coherent streaking artifacts. Zhao et al. [52] proposed a GAN-based sinogram inpainting network, which achieved unsupervised training in a sinogram-image-sinogram closed loop. Zhao et al. [53] also proposed a two-stage method, firstly they use an interpolating convolutional network to obtain the full-view projection data, then use GAN to output high-quality CT images. In 2019, Lee et al. [54] proposed a deep learning model based on fully convolutional network and wavelet transform. In the latest research, Zhang et al. [55] proposed an end-to-end hybrid domain CNN (hdNet), which consists of a CNN operating in the sinogram domain, a domain transformation operation, and a CNN operating in the image domain.

However, we cannot help but notice that, when it comes to image restoration, all the methods above merely focus on a single CT image while neglecting the solid fact that the scanned object are often spatially continuous. On account of that, these obtained consecutive CT images are always highly correlative, which leads to copious spatial information hidden between them that is still largely left to be explored. Consequently, we propose a novel two-step cascaded model in the second stage which concentrates on groundbreakingly utilizing the strong spatial correlation between consecutive CT images. So as to break the limit of feature extraction in the two-dimensional space and dig deep into the three-dimensional spatial neighborhood.

These two domains are also combined in our method to amalgamate their respective strengths for high-quality CT reconstruction results, which leads to our proposed three-stage structure. Specifically, we firstly conduct data completion in the Radon domain to acquire the full-view CT data, and then reconstruct it into images through FBP. Subsequently, image restoration and artifacts removal are accomplished in a “coarse-to-fine” [56] manner with the combination of stage two and stage three.

It is also worth mentioning that, unlike other current prevailing limited-view CT reconstruction methods [39], we adopt FBP [16] (and implement it on GPUs) instead of SART-TV [13] to speed up the overall procedure. Besides, since our method actually consists of fully convolutional networks, it does not limit the resolution of input images, thus can be well generalized to various datasets. In our experiments, we compare our algorithm with other methods under four sorts of limited-view CT data, exhibiting its prominent performance and robustness.

The organization of this paper is as follows, Section 2 presents our proposed method in detail, Section 3 exhibits the experimental results and corresponding discussion, and conclusion is stated in Section 4.

## 2. Methods

In this work, we propose a hybrid-domain limited-view CT reconstruction method, and its overall three-stage structure is shown in Figure 1. In the first stage, after the limited-view Radon data is preprocessed, we fed it into the Adversarial Autoencoder (AAE) established for data restoration, so as to acquire high-quality full-view Radon data, which is then transformed into images through FBP. In stage two, these CT images are concatenated into groups and then sent into our proposed Spatial Adversarial Autoencoder (Spatial-AAE) to perform image inpainting based on strong spatial correlation between consecutive CT images, which can manage to eliminate almost all the artifacts from the original limited-view CT images. However, we notice that the image texture of these restored CT images is still not precise enough compared to the ground truth CT images. Therefore, utilizing the idea of “coarse-to-fine” [56,57,58,59] in deep learning, we establish the Refine Adversarial Autoencoder (Refine-AAE) in the third stage to refine the image texture in patches, and eventually obtain the ideal high-quality CT images which are not only artifact-free, but also have fine image texture.

### 2.1. Preliminaries and Discussion

#### 2.1.1. How to Maximize the Limited Prior Information through Data Preprocessing

In order to obtain more valuable data from the limited prior information, we refer to [38] and adopt the data preprocessing method shown in Figure 2. For the limited-view Radon data $R_{lv}$, we first convert it into the image $I_{recon}$ through inverse radon transformation, and then adopt Radon transformation to transform $I_{recon}$ into the full-view Radon data $R_{fv}$. Subsequently, $R_{fv}$ is cropped for preliminary completion of the missing part in $R_{lv}$, so as to obtain the merged full-view Radon data $R_{merge}$. In this way, we manage to efficaciously utilize the existing data for better restoration results, which is proved in our experimental results from Section 3.

#### 2.1.2. How Does Spatial Correlation Help Remove Artifacts

As we mentioned above, since the scanned objects are always spatially continuous, the consecutive CT images obtained from them also have strong spatial coherence. Consequently, these continuous CT images can be regarded as successive frames from a video clip which have been proved to contain much more information than a single still image [60,61,62,63,64,65,66,67,68,69]. Specifically, the high correlation within the sequence of images helps remove artifacts from two perspectives. In the first place, it expands the search regions from the two-dimensional image neighborhoods to the three-dimensional spatial neighborhoods, thereby providing additional information which can be used to restore the reference image. Secondly, utilizing the consecutive CT images can be beneficial to remove artifacts as the residual error in each adjacent image is correlated.

Based on the analysis above, we notice the similarity between the task of artifact removal between successive images and the task of video denoising. Due to this similarity and the lack of relevant deep learning-based 3D CT reconstruction algorithms, we investigate lots of current prevailing research works in video denoising [60,61,62,63,64,65,66,67,68,69], and find out that these state-of-the-art methods give great prominence to motion estimation due to the strong redundancy along the motion trajectories. Therefore, we need a structure that can not only look into the three-dimensional spatial neighborhood, but can also conduct motion estimation between these consecutive images, so as to productively remove artifacts from limited-view CT images.

### 2.2. Overall Structure

#### 2.2.1. Stage One: Data Restoration in the Radon Domain

In this stage, we propose an AAE as shown in Figure 3, which is composed of an autoencoder and a discriminator. The parametric architecture of the autoencoder can be seen from Table 1, it incorporates an encoder and a decoder that are highly symmetrical. In the encoder, each building block (refers to Figure 4) extracts representative features and is followed a Maxpool Layer that conducts downsampling. Each downsampling here will halve the height and width of the activation map and double the number of channels, and the IC and OC stand for the number of input channels and output channels of these building blocks and layers. After obtaining the high-level semantic features from this encoder, we establish a decoder for image texture restoration. Transposed convolution is adopted here for feature upsampling with its stride and kernel size both equal to 2, each upsampling here will double the height and width of the activation map and halve the number of channels.

Besides, skip connections [40] are added between feature maps with the same resolution in the encoder and decoder. In the encoder, in order to acquire high-level semantic features, we conduct multiple downsampling which leads to the final feature map with a relatively low resolution, and makes it difficult for the decoder to restore the image texture. Thus, we need to utilize skip connections that can incorporate low-level features from the encoder which can help accurately precise image inpainting. It has been proved that, this sort of multi-scale, U-Net-like architectures can be well applied to medical image processing.

As for the discriminator, its structure is almost the same as the encoder above, except that its Block5 has three layers whose OCs are 512, 64 and 1 respectively. The output of Block5 is then flattened and fed into sigmoid function for probability prediction, which we average to get the final output that represents the input image’s probability to be a real image. This discriminator is added to strengthen the model’s ability to restore the detailed texture of images.

#### 2.2.2. Stage Two: Image Restoration Based on Spatial Correlation

After data completion in the Radon domain, we manage to mitigate the severe image artifacts to a certain extent (the specific visualized result can be seen from Figure 10 in Section 3). Nevertheless, the reconstruction result still needs to be further restored to thoroughly eliminate the artifacts and present the image texture. Therefore, we need an architecture that can effectively utilize the existing information to restore these CT images. As we mentioned above, almost all the current prevailing methods merely concentrate on a single CT images while ignoring the abundant spatial information between these consecutive CT images. Therefore, in this stage, we need to establish a model that can make full use of the spatial correlation. Recalling the discussion in II.A, we learn that this model should be capable digging into the three-dimensional spatial neighborhood and capturing motion between the continuous CT images.

Generally, an explicit motion estimation stage would have a relatively large memory cost, which may cause certain obstacles to its application. However, the two-step cascaded architecture in [70] appears to inherently embed the motion of objects with high efficiency. Enlightened by this, we establish the Spatial Adversarial Autoencoder that consists of the Spatial Autoencoder and the discriminator (its structure is the same as it is in stage one), the overall structure of the Spatial-AAE can be seen from Figure 5.

The input of the Spatial Autoencoder is five consecutive CT images $S=\{s_{i-2},s_{i-1},s_{i},$$s_{i+1},s_{i+2}\}$, *S* is divided into three sets of image sequences $S_{1}=\left\{s_{i-2},s_{i-1},s_{i}\right\},S_{2}=\left\{s_{i-1},s_{i},s_{i+1}\right\}$ and $S_{3}=\left\{s_{i},s_{i+1},s_{i+2}\right\}$. Then, they are fed into the AE block respectively, and their output is concatenated as $S^{\prime }$, which is sent into the AE block (trained separately from the AE block in the first step) to obtain the final restoration result. This whole structure can be expressed as Equation (Equation 1), where *F* represents the Spatial Autoencoder and *G* stands for the AE block. The specific details of the AE block can be seen from Table 1.
(1)$$s_{i}^{\prime \prime }=F\left(S\right)=G\left(G\left(S_{1}\right),G\left(S_{2}\right),G\left(S_{3}\right)\right)$$

#### 2.2.3. Stage 3: Image Refining on Patches

After the above two stages of hybrid-domain restoration, the limited-view CT reconstruction result can reach a relatively satisfying degree (the specific visualized result can be seen from Figure 10 in Section 3). Nevertheless, the image texture is still not precise enough compared to the ground truth CT images, thereby need to be further refined. In this stage, we utilize the idea of “coarse to fine” in deep learning, and propose the Refine Adversarial Autoencoder to refine the coarse results obtained from the second stage. The overall structure of the Refine-AAE is shown in Figure 6, which is composed of the Refine Autoencoder and the discriminator (its structure is the same as it is in stage one). More importantly, we crop the input image into four patches of the same size and adjust them to the same pattern, so that it would be easier for the model to learn this mapping from this fixed pattern.

Specifically, given the input image $I_{input}$, the Refine Autoencoder firstly divides it into four patches, then use horizontal and vertical flip to convert them into the same pattern. After this, the patches are concatenated into sequence $\left\{I_{p1},I_{p2},I_{p3},I_{p4},\right\}$ and fed into our AE block for texture refinement. we obtain the prediction result $\left\{I_{p1}^{{}^{\prime }},I_{p2}^{{}^{\prime }},I_{p3}^{{}^{\prime }},I_{p4}^{{}^{\prime }},\right\}$ and integrate it into $I_{pred}$, then it is combined with the ground truth CT image $I_{GT}$ into pair for discriminator’s judgment.

### 2.3. Network Training

All these stages are optimized separately with Adam [71] (set the learning rate to $1\times 10^{-4}$ at the beginning), and we adopt the multi-loss function for all the autoencoders in these neural networks, the loss function is composed of $l_{MSE}$, $l_{Adv}$ and $l_{Reg}$ with their respective hyperparameters $\alpha_{1}$, $\alpha_{2}$ and $\alpha_{3}$ set to 1, $1\times 10^{-3}$, and $2\times 10^{-8}$ respectively during training.
(2)$$l_{AE}=\alpha_{1}l_{MSE}+\alpha_{2}l_{Adv}+\alpha_{3}l_{Reg}$$

In Equation (Equation 2), $l_{MSE}$ calculates the mean square error between the prediction result and its corresponding ground truth, this loss function is widely used in image inpainting because it can provide an intuitive evaluation for prediction results. The expression of $l_{MSE}$ is shown in Equation (Equation 3).
(3)$$l_{MSE}=\frac{1}{W\times H}\sum_{x=1}^{W}\sum_{y=1}^{H}{\left(I_{x,y}^{GT}-G_{AE}{\left(I^{Input}\right)}_{x,y}\right)}^{2}$$

In Equation (Equation 3), *W* and *H* are the width and height of the input image respectively, $I^{Input}$ and $I^{GT}$ stand for the input image and its corresponding ground truth, function $G_{AE}$ represents the autoencoder.

In Equation (Equation 2), $l_{Adv}$ calculates the adversarial loss, which can be minimized to make the prediction result as close to the real data distribution as possible. Its expression is shown in Equation (Equation 4).
(4)$$l_{Adv}=1-D\left(G_{AE}\left(I^{Input}\right)\right)$$
where $I^{Input}$ stands for the input image, function *D* and $G_{AE}$ represent the discriminator and the autoencoder respectively.

In Equation (Equation 2), $l_{Reg}$ plays the role of a regularizer in our multi-loss function. As we know, noises are harmful to image inpainting, thereby we need a regularizer to smooth the image while preventing overfitting. Since TV Loss is widely used in image analysis, which can effectively reduce the variation between adjacent pixels, and the expression is shown in Equation (Equation 5).
(5)$$l_{Reg}=\frac{1}{W\times H}\sum_{x=1}^{W}\sum_{y=1}^{H}∥\nabla G_{AE}\left(I_{x,y}^{Input}\right)∥$$
where *W* and *H* stand for the width and height of the input image, $∥\cdot ∥$ acquires the norm, ∇ calculates the gradient, function $G_{AE}$ stands for the autoencoder, $I^{Input}$ represents the input image.

As for the optimization of the discriminators in these stages, we minimize the loss function below to make the discriminators better distinguish between real and fake input images. The loss function $l_{Dis}$ can be seen from Equation (Equation 6).
(6)$$l_{Dis}=1-D\left(I^{GT}\right)+D\left(G_{AE}\left(I^{Input}\right)\right)$$
where function *D* and $G_{AE}$ represent the discriminator and the autoencoder, $I^{Input}$ and $I^{GT}$ are the input image and its corresponding ground truth. The discriminator outputs a scalar between 0 to 1 that stands for the probability of the input image being real. Therefore, minimizing $1-D(I^{GT})$ and $D\left(G_{AE}\left(I^{Input}\right)\right)$ enables the discriminator to distinguish fake images (prediction results of the autoencoders) from all input images.

## 3. Experiment

We adopt the LIDC-IDRI [72] dataset and divide its 1018 cases (approximately 240,000 DCM files) into train set, validation set and test set according to the ratio of 1:1:3, and the amount of data is relatively large enough for us to train our models from scratch. We process these DCM files, read them into NumPy arrays, adopt normalization to ensure all data are scaled to the same range and create four sorts of limited-view CT data with varying degree of artifacts (the corresponding full-view CT data has 180 projection views). A geometry representative of a 2D parallel-beam CT scanner setup was used, and the sinogram was simulated by forward projecting the clinical images. The resolution of the CT image was 512 × 512 pixels, and each view of simulated sinogram was modeled with 512 bins on a 1D detector. In this section, we first conduct ablation studies to prove the rationality of our structural design, and then compare our method with other current methods under various limited-view CT data, exhibiting its remarkable performance and robustness. In addition, if not specifically mentioned, all the experiments in III.A are conducted with the limited-view CT data which lacks the post 60 projection views.

### 3.1. Ablation Study

#### 3.1.1. Data Preprocessing

In our data preprocessing, to make full use of the finite prior information, we preliminarily complement the missing projection views of the original limited-view CT data (refers to Figure 2). Therefore, we conduct an experiment to see how much the additional information can help improve restoration results in the first stage. In this experiment, we feed the limited-view Radon data and the merged full-view Radon data into the AAE in stage one respectively, and then compare their restoration results with the corresponding ground truth, which can be seen in Table 2 and Figure 7. OR and MR stands for the original limited-view Radon data and the merged Radon data, ROR and RMR represents the restored OR and the restored MR from stage one.

We can see from the quantitative and visualized experimental results that, MR can obtain significantly better restoration outcome, and its image texture is obviously closer to the ground truth, proving the effectiveness of our data preprocessing method.

#### 3.1.2. The Role of Our Discriminator

We employ our proposed discriminators in all three stages, aiming to obtain finer restoration results. Thus, we feed the merged Radon data into these two models respectively: (1) Merely the autoencoder (refers to Table 1); (2) Combination of the autoencoder and the discriminator, its quantitative and visualized experimental results can be seen from Table 3 and Figure 8.

We notice that the image texture of the rear 60 projection views in Figure 8c is obviously finer than Figure 8b. Also, the restoration result of structure (2) is pretty close to the ground truth as it is shown in Figure 8. Thereby, we can safely arrive at the conclusion that, the discriminator plays an important role in improving the restoration results.

#### 3.1.3. The Two-Step Cascaded Architecture: Spatial-AAE

Since the Spatial-AAE is proposed to efficaciously utilize the spatial correlation between consecutive CT images through the cascaded two-step architecture, which can manage to dig into the three-dimensional neighborhood and inherently embeds the motion of objects. To verify the effectiveness of this structural design, we carry out an experiment with reference to [70] to prove this view. In stage two, instead of feeding five successive images into Spatial-AAE, we send them directly into a single AE block (SAE) that is not capable of conduct implicit motion estimation. The experimental results can be seen from Table 4, the discriminator is also added to the SAE to ensure fairness.

As we know, the AAE does not own this built-in cascade structure like Spatial-AAE to implicitly exploit the spatial correlation, it suffers from a great drop in PSNR and SSIM. This also allows us to further think about the characteristics and advantages of the Spatial-AAE architecture. Compared with AAE, Spatial-AAE adopts a two-step cascade model to implicitly perform motion estimation, and we also learned that such a process can effectively learn from residual information in consecutive images to provide additional extra prior information for restoration, thus improving the overall restoration performance to a certain extent. Besides, motion estimation needs to consume a large amount of additional computing resources in general, while such a two-step implicit motion estimation structure can manage to effectively avoids this, also create a deeper neural network to enhance the overall repair ability of the model.

On account of these, we can safely arrive at the conclusion that, this sort of architecture can help effectively improve the restoration results.

#### 3.1.4. Refine the Image Texture in Patches

In the third stage, the input image is divided and concatenated, then sent to the Refine-AAE for finer restoration. We believe that refining the image texture in patches makes it easier for the model to learn the mapping and obtain better restoration results. In addition, we want to verify the effect of different patch interception methods on the final restoration results.

To prove the above points, we design an experiment that feeds these four types of data into the model in stage three: Method 1, randomly crop four patches (size 256 × 256) from the input image (size 512 × 512); Method 2, crop the four corners out of the input image; Method 3, crop the four corners out of the input image, and then adjust them into the same pattern through different flipping method; Method 4, no cropping. Diagrams of the first three patch interception methods are shown in Figure 9, and the corresponding quantitative restoration results can be seen from Table 5.

We can see that, if the patches are randomly cropped, it would to lead to a relatively poor restoration result since the pattern of input patches are complicated. However, when we adopt corner crop (with or w/o flip), its outcome exceeds method 4 due to its fixed pattern which may be easier for neural networks to learn. In addition, it is worth mentioning that method 2 has the best performance, even surpassing method 3, which particularly employs flips to adjust patches to the same pattern. It seems that the non-flip in method 2 works in the form of data augmentation, thereby improving the restoration results.

#### 3.1.5. Refine the Image Texture in Patches

We previously delved into the precise design of the overall architecture, which is divided into three successive stages. Here, to demonstrate their effectiveness, quantitative and intuitive experimental results are shown in Table 6 and Figure 10.

As can be seen from Figure 10b, the first stage manages to alleviate the severe image artifacts, while there still remains some minor image impairments that require further improvement. Fortunately, after two stages of hybrid-domain restoration, the limited-view CT reconstruction result (refers to Figure 10c) can reach a relatively satisfying degree with no apparent artifacts. In addition, we adopt stage three to further improve the experimental results by a relatively small margin, which can also be verified by Table 6.

### 3.2. Methods Comparison

After verifying the rationality of our structural design, and then compare our method with other current methods under various limited-view CT data. The methods include: (1) Analytical reconstruction algorithm FBP; (2) Iterative reconstruction algorithm SART combined with TV regularization; (3) Image inpainting with U-Net, after reconstructing the limited-view Radon data into images through FBP, adopt U-Net for image restoration; (4) Sinogram inpainting with U-Net, first adopt U-Net to complement the limited-view Radon data, then reconstruct it to images through FBP. In addition, in order to testify the effect of merging Radon data (MR), we implement these methods on the two sorts of input data: (1) the original limited-view Radon data; (2) the merged full-view Radon data (data preprocessing). For limited-view CT data which lack the post 60 projection views, the quantitative and intuitive restoration results of the above methods are shown in Table 7 and Figure 11. In Table 7, we evaluate all these methods’ performance on the test set with their mean and standard deviation (std) to provide additional measurement for stability.

From the quantitative results above, we can see that MR manages to bring additional information for every method, thereby improving their performance by different margin. Besides, restoration using U-Net, which is known to be very effective in processing biomedical images, appears to be less useful in the Radon domain. In this case, our method combines these two domains to take advantage of their respective strengths, and finally obtain a extraordinary outcome that achieves the PSNR of 40.209 and the SSIM of 0.943, while exhibiting its stability on various limited-view data. More importantly, it not only improves the image quality by a large margin, but also realizes the precise restoration of image texture that few methods can achieve. To further demonstrate this, we calculate the corresponding error maps (refers to Figure 12), which exhibits the difference between the restoration results and the ground truth CT images.

Moreover, to testify the robustness of these methods, we implement them on three sorts of limited-view data that have more serious artifacts in their imaging. Including (1) limited-view CT data that lacks the middle 60 projection views; (2) limited-view CT data that lacks the middle 90 projection views; (3) limited-view CT data that lacks the middle 120 projection views, and their corresponding full-view data has 180 projection views. The experimental results can be seen from Figure 13 and Table 8.

The performance of these methods has been greatly affected by the increasing information loss. Our method, however, demonstrates its outstanding robustness and still exceeds other methods by a large margin under varying degrees of damaged data.

## 4. Conclusions

In order to obtain the ideal high-quality restoration results from the limited-view CT images that contains severe artifacts, we propose a hybrid-domain structure that efficaciously utilizes the spatial information between consecutive CT images, and utilizes the idea of “coarse to fine” to refine the image texture.

In the first stage, we establish an adversarial autoencoder to preliminarily complement the original limited-view Radon data. After converting the obtained full-view Radon data into images through FBP, and feed them into our proposed Spatial-AAE in stage two for artifacts removal based on spatial information. By now, we have managed to thoroughly eliminate the severe artifacts from the original limited-view CT images, while the image texture still needs to be further refined. Therefore, in the third stage, we propose the Refine-AAE to refine the image in the form of patches, so as to achieve the accurate restoration of the image texture.

For limited-view Radon data that lacks the rear 60 projection views, our method can increase its PSNR to 40.209, and SSIM to 0.943, not only largely improve the image quality compared to other current methods, but also precisely present the image texture. At the same time, our method can be well applied to other sorts of limited-view CT data with more serious artifacts in their imaging, demonstrating its remarkable robustness. 

## Figures and Tables

**Figure 1 sensors-22-01446-f001:**
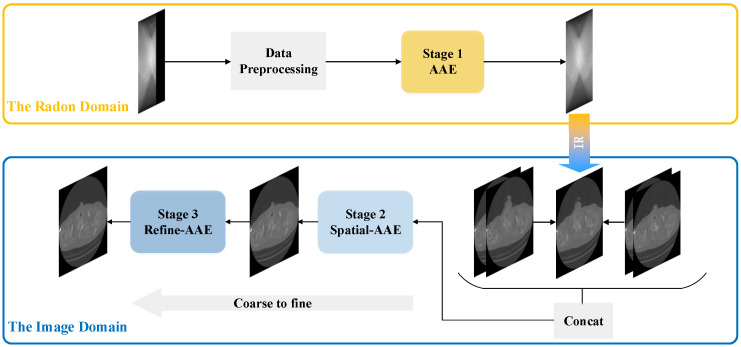
The overall architecture of our proposed method.

**Figure 2 sensors-22-01446-f002:**
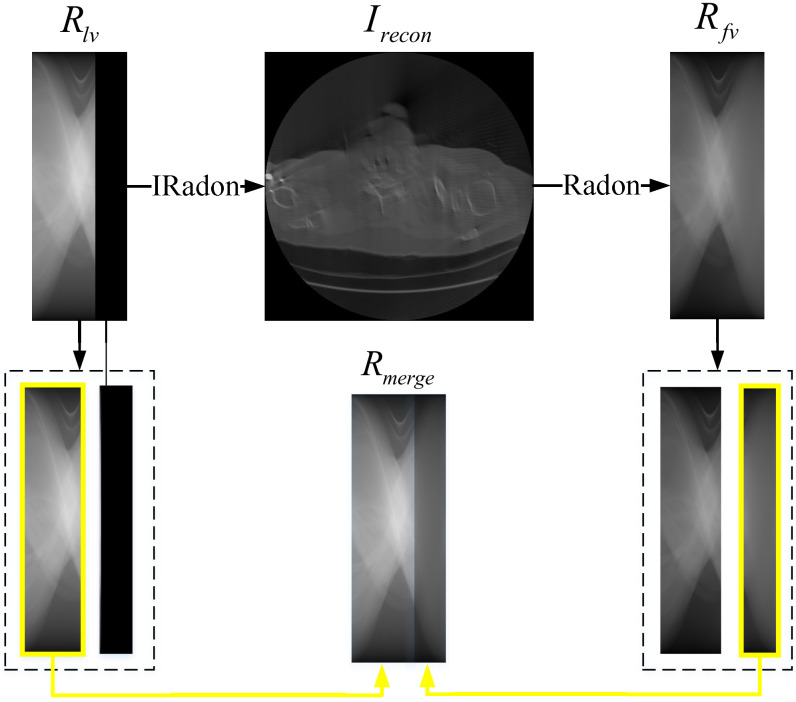
Workflow of data preprocessing.

**Figure 3 sensors-22-01446-f003:**
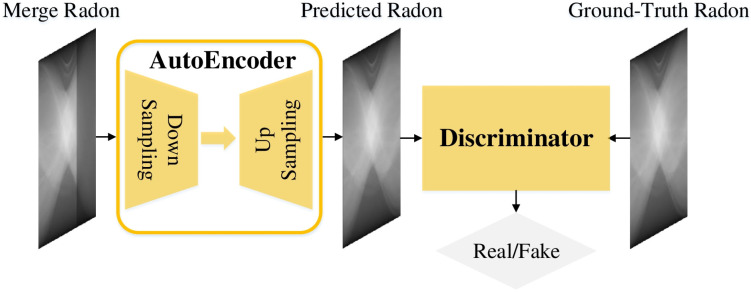
The overall architecture of our proposed AAE in stage one.

**Figure 4 sensors-22-01446-f004:**
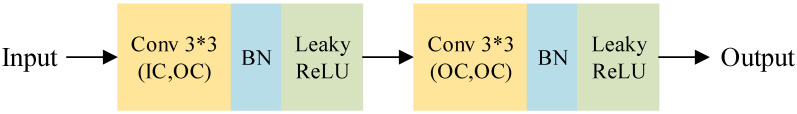
The diagram of the building block in AAE.

**Figure 5 sensors-22-01446-f005:**
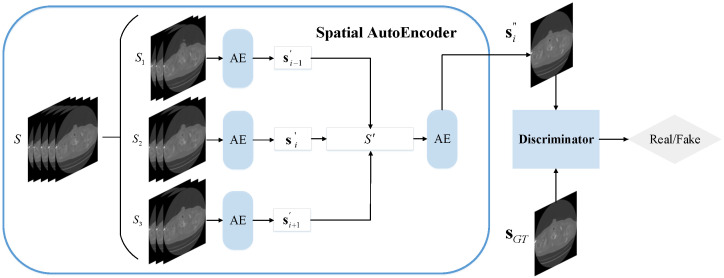
The overall architecture of our proposed Spatial-AAE in stage two.

**Figure 6 sensors-22-01446-f006:**
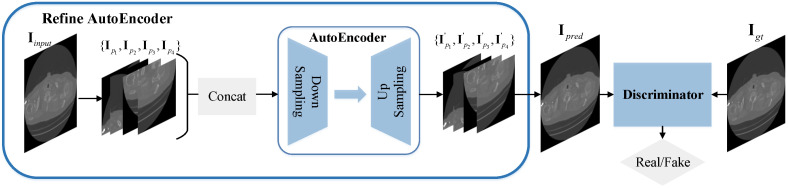
The overall architecture of our proposed Refine-AAE model in stage three.

**Figure 7 sensors-22-01446-f007:**
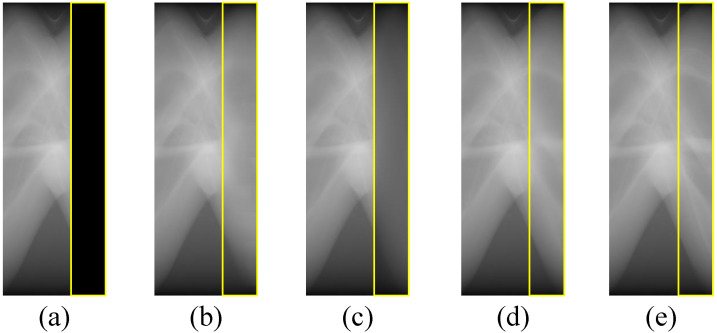
Visualized results obtained from different data preprocessing methods, (**a**) is the directly cut Radon data; (**b**) is the restored result of (**a**); (**c**) is the fused Radon data; (**d**) is the restored result of (**c**); (**e**) is the Radon ground truth.

**Figure 8 sensors-22-01446-f008:**
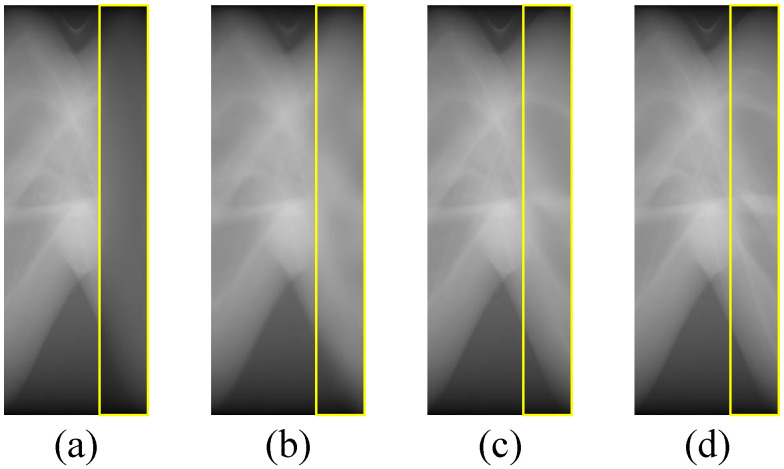
Visualized restoration results obtained from different data preprocessing methods, (**a**) is the input; (**b**) is the restoration result of structure (1); (**c**) is the restoration result of structure (2); (**d**) is the ground truth.

**Figure 9 sensors-22-01446-f009:**
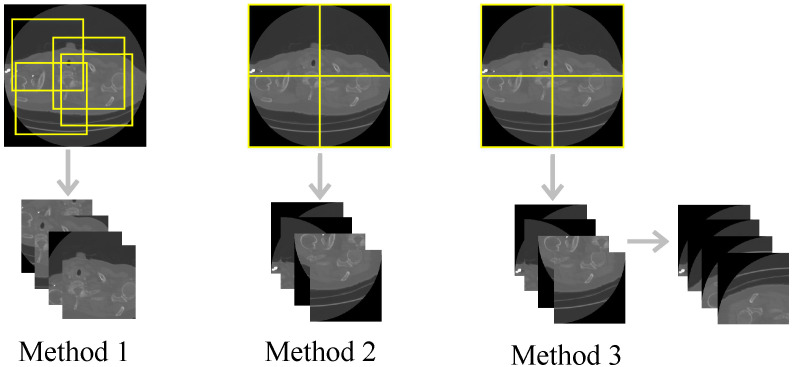
Methods of cropping patches in stage three.

**Figure 10 sensors-22-01446-f010:**
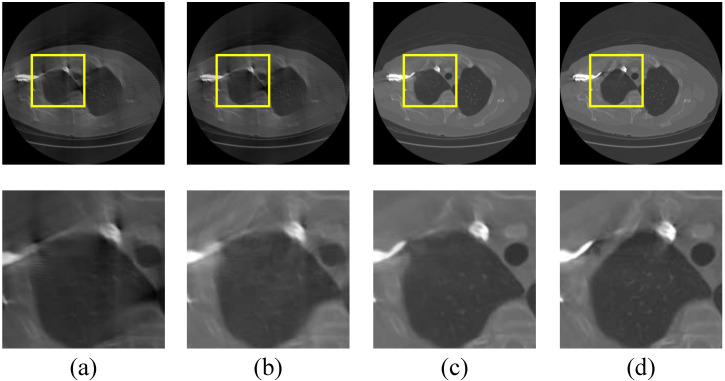
(**a**) is the original limited-view CT image, which lack the post 60 projection views (the corresponding full-view CT has 180 projection views); (**b**) is the CT reconstruction result of the first stage; (**c**) is the CT reconstruction result of the second stage; (**d**) is the ground truth CT image.

**Figure 11 sensors-22-01446-f011:**
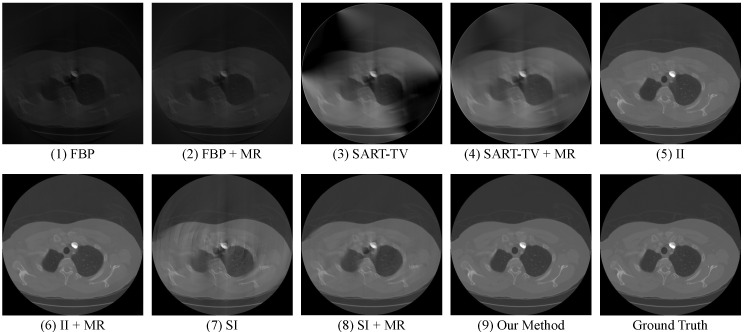
Visualized restoration results of various methods.

**Figure 12 sensors-22-01446-f012:**
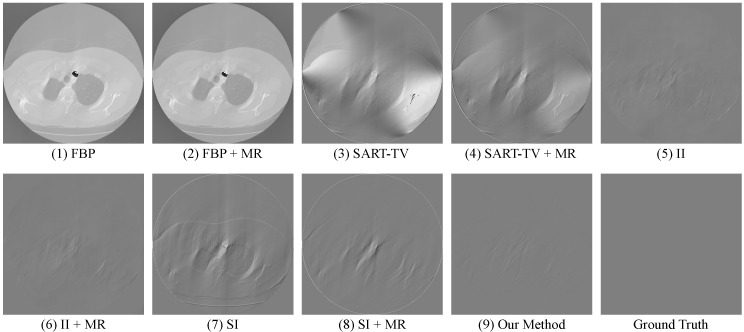
Error maps of the restoration results obtained by various methods.

**Figure 13 sensors-22-01446-f013:**
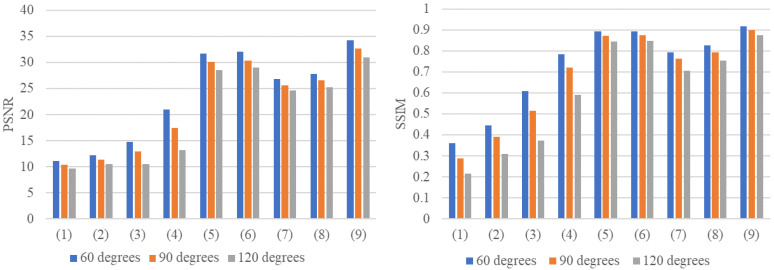
Histograms of different algorithms applied to different data preprocessing methods on different input data.

**Table 1 sensors-22-01446-t001:** Parametric Structure of the AAE.

Layer	*IC*	*OC*	Stride	Input Size	Output Size
Block1	1	32	1	192 × 512	192 × 512
Pool1	32	32	2	192 × 512	96 × 256
Block2	32	64	1	96 × 256	96 × 256
Pool2	64	64	2	96 × 256	48 × 128
Block3	64	128	1	48 × 128	48 × 128
Pool3	128	128	2	48 × 128	24 × 64
Block4	128	256	1	24 × 64	24 × 64
Pool4	256	256	2	24 × 64	12 × 32
Block5	256	512	1	12 × 32	12 × 32
Up_Conv6	512	256	2	12 × 32	24 × 64
Block6	256 + 256 (Concat)	256	1	24 × 64	24 × 64
Up_Conv7	256	128	2	24 × 64	48 × 128
Block7	128 + 128 (Concat)	128	1	48 × 128	48 × 128
Up_Conv8	128	64	2	48 × 128	96 × 256
Block8	64 + 64 (Concat)	64	1	96 × 256	96 × 256
Up_Conv9	64	32	2	96 × 256	192 × 512
Conv9_1	32 + 32 (Concat)	32	1	192 × 512	192 × 512
Conv9_2	32	12	1	192 × 512	192 × 512
Conv9_3	12	1	1	192 × 512	192 × 512

**Table 2 sensors-22-01446-t002:** Restoration results of OR and MR.

	OR	MR	ROR	RMR
PSNR	8.714	18.196	38.549	48.181
SSIM	0.656	0.936	0.987	0.995

**Table 3 sensors-22-01446-t003:** Restoration results of different structures.

	AE	AE + D
PSNR	40.129	48.181
SSIM	0.983	0.995

**Table 4 sensors-22-01446-t004:** SAE vs. Spatial-AAE.

	AAE	Spatial-AAE
PSNR	37.384	39.646
SSIM	0.929	0.940

**Table 5 sensors-22-01446-t005:** Restoration results of different patch interception methods.

	Random Crop	Corner Crop	Corner Crop + Flip	No Cropping
PSNR	39.863	40.209	40.06	39.948
SSIM	0.941	0.943	0.942	0.941

**Table 6 sensors-22-01446-t006:** Restoration results of different patch interception methods.

	Original Input	Stage One’s Output	Stage Two’s Output	Final Output
PSNR	22.417	28.960	39.646	40.209
SSIM	0.812	0.859	0.940	0.943

**Table 7 sensors-22-01446-t007:** Methods Comparison.

Algorithms	PSNR (Mean ± Std)	SSIM (Mean ± Std)
(1) FBP	11.272 ± 0.917	0.364 ± 0.017
(2) FBP+MR	12.354 ± 0.811	0.452 ± 0.015
(3) SART-TV	14.727 ± 0.824	0.635 ± 0.021
(4) SART-TV+MR	21.518 ± 0.729	0.807 ± 0.019
(5) Image Inpainting (II)	35.566 ± 2.283	0.916 ± 0.047
(6) Image Inpainting + MR	36.388 ± 2.106	0.927 ± 0.047
(7) Sinogram Inpainting (SI)	27.345 ± 2.476	0.800 ± 0.014
(8) Sinogram Inpainting + MR	28.960 ± 2.461	0.859 ± 0.013
(9) Ours	40.209 ± 1.325	0.943 ± 0.015

**Table 8 sensors-22-01446-t008:** Restoration results of various algorithms for limited-view data with varying degrees of artifacts.

	CUT-MID-60		CUT-MID-90		CUT-MID-120	
Algorithms	PSNR	SSIM	PSNR	SSIM	PSNR	SSIM
(1) FBP	11.131	0.362	10.350	0.289	9.636	0.217
(2) FBP + MR	12.182	0.446	11.432	0.391	10.525	0.309
(3) SART-TV	14.758	0.610	12.945	0.515	10.492	0.372
(4) SART-TV + MR	21.036	0.784	17.523	0.722	13.166	0.592
(5) Image Inpainting (II)	31.717	0.895	30.157	0.873	28.507	0.846
(6) Image Inpainting + MR	32.031	0.895	30.422	0.876	28.999	0.849
(7) Sinogram Inpainting (SI)	26.834	0.793	25.673	0.763	24.606	0.705
(8) Sinogram Inpainting + MR	27.789	0.828	26.582	0.795	25.210	0.755
(9) Ours	34.248	0.919	32.624	0.900	30.975	0.876

## Data Availability

Not applicable.

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
