# Peer review of "A Limited-View CT Reconstruction Framework Based on Hybrid Domains and Spatial Correlation"

_sensors, 2022, doi:10.3390/s22041446_

Round 1
Reviewer 1 Report
From my perspective, the paper-work is well structured and propose a new hybrid-domain structure (convolutional networks).
Main steps:
-data completion in the Radon domain
-transform full-view Radon data=>images through FBP.
-spatial correlation between CT images
-refine the image texture
- The architecture of the proposed method is in Figure 1.
- The workflow of data preprocessing is presented in Figure 2.
- The other steps are illustrated in the following Figures.
- The Parameters of the AAE- Table 1
-Testing data- LIDC-IDRI [73] dataset divided 1018 cases=>40,000 DCM files training data-set
-Comparison is made between: (1) Analytical reconstruction algorithm FBP; (2) Iterative reconstruction algorithm SART + TV regularization; (3) Image inpainting with U-Net (4) Sinogram inpainting with U-Net
Observations:
pag.8: 236 and similar ones
\noindent where ....
Some advantages and few (indirectly written) disadvantages of the new method are included.
Optional: Introduction could include also recent bio-inspired computing with different operators for Medical Images including CT ones e.g. ant algorithm, artificial bee colony optimisation, fire-fly optimization.
Author Response
Thank you very much for your reminder. We think this is a remarkable inspiration and deserves more in-depth and specific research. Unfortunately, these optimization algorithms do not quite match our main topic in this paper. We suppose including them in this paper may create some major conflicts. However, we are looking forward to create some exciting work about bio-inspired computing optimization algorithms in the future.
Reviewer 2 Report
The submitted paper proposes a Low-Dose CT reconstruction framework massively exploiting neural networks in three inner steps.
With respect to the state of art, the main contribution lies in the exploitation of 3D-like correlations among the CT slices, in the second step.
Indeed, so far, few works have already explored the importance of spatial correlations and it makes the paper of actual interest.
In the following, my suggestions to enforce the paper and the weaknesses I have found in the submitted work.
(1) the paper is not well organized, hence I suggest a deep revision to its organization. For instance, the experiments reported in Fig 5 and 7 may be included in section 3.
(2) The cited papers 61-70 (inspiring your work) address video denoising whereas you do not cite nor discuss papers about deep learning-based 3D CT reconstructions at all. Please, consider adding this kind of works and how they differ from 61-70. In particular, I claim that noise (on video and on images as well) is space invariant mostly, hence learning its statistics is very different than learning streaking artifacts.
For this reason, it could be interesting (and consistent with the novelty of your work) to compare your results with such approaches.
(3) The experiments may be better presented. Some points:
a) Have you used parallel- or cone-beam simulations? Which is your scanning range?
b) Do the Tables refer to one image or the test set? In the case of mean values on the test set, please add a dispersion measure.
c) In Fig 11, Methods 1-2-3 are labels not assigned in the text.
d) In Fig 12, what does "intuitive" stand for?
e) Focusing on the methods for comparison: are they executed until convergence? How is it defined?
f) are the images reported in the same gray-scale range?
(4) In the experiments, I would expect an analysis to explore the benefits of the 3D spatial neighborhood on the final volumetric reconstructions.
Reviewer 3 Report
Overall comments:
The author described the method for limited- angle image reconstruction, but the
concept of inpainting network using autoencoders in the process has been mentioned in
the some articles during 2018-201919 years. The concept and methodology are not
novel enough, and the relationship between the inpainting point and the front and rear
pixels is not mentioned in the inpainting process.
1. The material in CT data set is the complete dataset. How to make limited-angle
data? Why reconstructed the limited-angle data and then appl ied the Radon
transform could be identified and removed on AAE and S-AAE?
2. The author did not explain why the CT images after reconciliation through S-
AAE will have better results than AAE? The physical principles are not fully
explained in the research.
3. The author used “coarse-to-fine pyramid model, to recover the missing data part.
Why can it work on missing region f or recoverery? Is there any evidence to
support this point?
Overall, the authors need to explain in more detail and not just show the architecture
structure.
Major revision is needed.
Round 2
Reviewer 2 Report
The authors have clearly answered my questions point-by-point making a few minor revisions to the submitted text.
I only highlight that the overall presentation could be reorganized and improved, for simpler readability and better fitting the standards of an academic journal.
For instance, in section "Experiments", the authors could add details (as some of the answers to my questions) about the CT geometry and how the images are reported. Focusing on the values reported in the tables, I claim that it is not specified whether they are mean (or median) values relative to the whole test set. In my opinion, the STD values could highlight the stability of the proposed DL-based approach (however, not mandatory to report them in Tables with the popular notation "mean \pm std").
At last, the submitted paper has missing inner references and citations.
Reviewer 3 Report
This manuscript version-2 has revised and add lots of descriptions. But the figure, table and references super-connections into the main text are disappeared. There are two comments I suggest the author can consider and modify.
-
- The restoration in this article, the deep learning model play an important role in extrapolation based inpainting. We suggest the author can find another article in Sensor 2018, 18, p.4458. The referenced article is use interpolation based inpainting to improve because the interpolation based let the training inpainting region would be localized, but extrapolation based may cause region diverged.
- In 3.1.3, line 309-311: author mentioned that "Compared with AAE, Spatial-AAE adopts a two-step cascade model to implicitly perform motion estimation, and we also learned that such a process can effectively learn from residual information in consecutive images to provide additional extra prior information for restoration, thus improving the overall restoration performance to a certain extent."
Is the CT sinogram data motion? Why the author regard as the motion estimation need to be concern? I have some confusing.
